# Improvement in Physicochemical and Functional Properties of Insoluble Dietary Fiber from Rice Bran Treated with Different Processing Methods

**DOI:** 10.3390/foods14173116

**Published:** 2025-09-05

**Authors:** Yanxia Chen, Qin Ma, Fei Huang, Xuchao Jia, Lihong Dong, Dong Liu, Mingwei Zhang, Ruifen Zhang

**Affiliations:** 1Sericultural & Agri-Food Research Institute Guangdong Academy of Agricultural Sciences, Key Laboratory of Functional Foods, Ministry of Agriculture and Rural Affairs, Guangdong Key Laboratory of Agricultural Products Processing, Guangzhou 510610, China; 18016690595@163.com (Y.C.); qinma_gaas@163.com (Q.M.); hf1311@163.com (F.H.); jiaxuchao@gdaas.cn (X.J.); dolify@163.com (L.D.); 2School of Food and Drug, Shenzhen Polytechnic University, Shenzhen 518055, China; liudongsz@szpu.edu.cn; 3Food Laboratory of Zhongyuan, Luohe 462300, China

**Keywords:** rice bran, insoluble dietary fiber, fermentation, extrusion, functional properties

## Abstract

Rice bran represents an exceptional natural source of dietary fiber (DF), and its physicochemical properties and therapeutic potential are closely associated with its origin and processing methods. Herein, rice bran was subjected to extrusion, fermentation, and a combined treatment of fermentation and extrusion to explore the alternations in the structural, physicochemical, and functional properties of the resulting insoluble dietary fiber (IDF). All treatments induced substantial microstructural alterations in IDF, producing fiber matrices with enhanced porosity and looser architectures. The employed processing treatments significantly enhanced the functional properties of rice bran IDF over the unprocessed sample, with 1.37- to 1.78-fold increases in oil-holding capacity, 1.31- to 1.48-fold increases in cholesterol-adsorption capacity, 2.89- to 5.90-fold increases in α-amylase-inhibitory activity, and 2.41- to 3.70-fold increases in glucose-adsorption capacity. Among them, extrusion proved more effective than fermentation in enhancing the water-holding capacity, sodium cholate binding, and cholesterol-adsorption capacity of rice bran IDF. However, fermented rice bran-derived IDF exhibited the optimum α-amylase-inhibitory and glucose-absorption capacities among all employed IDF samples. These findings provide valuable insights for the development of rice bran-based functional foods with enhanced health benefits.

## 1. Introduction

Dietary fiber (DF), an indigestible carbohydrate from plant cell walls, is the seventh most important nutrient for humans [1,2]. DF can be divided into soluble (such as resistant oligosaccharides, gums, and pectin) and insoluble (such as lignin, cellulose, and hemicellulose) dietary fibers (SDF, IDF) [3], in which IDF accounts for approximately 60% of the total DF. DF is recognized for enhancing digestion, stimulating bowel movements, lowering blood glucose levels, and reducing the risk of coronary heart disease, diabetes, obesity, and certain cancers [4,5,6]. Whole grains, nuts, fruits, and vegetables are excellent sources of DF, with whole grains contributing approximately 50% of the total DF intake in Western countries [7].

As an integral component of whole-grain brown rice, the bran layer of brown rice is abundant in DF, with IDF accounting for approximately 90% of the total [8]. Research has confirmed the significant health benefits associated with cereal-derived DF, including promoting intestinal health, regulation of blood glucose and lipid metabolism, and enhanced satiety [9]. Recent studies have further revealed that cereal-derived IDF possesses physical and chemical as well as functional properties, such as water-holding capacity, oil-binding capacity, and glucose-adsorption capacity [10]. However, a high IDF content also results in poor palatability, which significantly restricts the utilization of brown rice bran in health foods [11,12].

Current approaches to enhancing whole-grain quality encompass physical, chemical, and biological methodologies [12]. Among these, physical and biological methods often gain favor due to their operational simplicity, reduced chemical residues, and environmental compatibility. Physical modification techniques like extrusion are widely employed due to their efficacy in altering texture and functional properties. Concurrently, biological approaches, particularly fermentation, leverage enzymatic or microbial action to modify substrates, potentially enhancing nutritional value, bioavailability, and sensory attributes [13,14]. Recent research highlights the effectiveness of both extrusion and fermentation in modifying DF from various whole-grain sources. Extrusion processing has been demonstrated to significantly promote the structural and physicochemical characteristics of cereal DF, often improving solubility, hydration properties, and overall functionality. Similarly, fermentation, employing specific microorganisms or enzymes, can modify the structure of bran DF, potentially reducing anti-nutritional factors, increasing SDF content, releasing bound bioactive compounds, and improving digestibility and palatability. Ren et al. found that through fermentation and enzymatic treatment, the glucose-adsorption capacity of DF was significantly enhanced (by 57.2–66.1%) [13,15,16,17]. These changes can enhance the health-promoting effects (such as prebiotic potential and antioxidant activity) and functional applications of cereal bran. Identical extrusion and fermentation treatments significantly increased the phenolic content in rice bran. The total phenolic content increased by 36.3% after extrusion, and by 71.6% after fermentation [18]. Previous studies have shown that both extrusion and fermentation with Rhizopus oryzae can promote the release of phenolic compounds from DF (including IDF), suggesting their effectiveness in modulating the structure and functional properties of the fiber. Extrusion processing can modify dietary fiber by altering physical parameters such as temperature and pressure, while fermentation with Rhizopus oryzae can change the crystallinity of cellulose and glycosidic bond linkages by releasing carbohydrate-active enzymes [19,20].

In this study, brown rice bran was processed using fermentation, extrusion, and their combination, and IDF samples from various processed brown rice brans were extracted to investigate the effect of different processing methods on structural, physicochemical, and functional characteristics of DF from brown rice. The results preliminarily revealed the underlying patterns of physical and biological processing in modifying the functional characteristics of rice bran IDF, thereby providing a practical foundation for the processing and application of cereal-based IDF.

## 2. Materials and Methods

### 2.1. Materials

Fresh rice bran was acquired from Guangdong Honeylong Agricultural Technology Co., Ltd. (Zhuhai, China) Defatted rice bran (DRB) was prepared by grinding the material, sieving it through an 80-mesh sieve, and then subjecting it to two 24 h shaking cycles with ten times its volume of petroleum ether. Heat-stable α-amylase (Termamyl SC, 120 KNU/g), protease (Alcalase 2.4 L, 2.4 AU/g), and amyloglucosidase (AMG 300 L, 300 AGU/g) were sourced from Novozymes (Tianjin, China) Biotechnology Co., Ltd. All other chemicals, solvents, and reagents used in the experiments were of analytical grade.

### 2.2. Preparation of Rice Bran Using Different Processing Methods

#### 2.2.1. Extruded DRB (EDRB)

DRB was treated with a DS32-II twin-screw extruder from Saixin Machinery Company Limited, based in Jinan, China, to produce extruded rice bran (EDRB). According to the conditions of the laboratory’s previous exploration [18], the extrusion parameters were set as follows: adjust the moisture content of DRB to 25%, and set the temperatures of the front, middle, and rear sections of the extrusion machine to 70, 98, and 134 °C with a screw speed of 300 r/min.

#### 2.2.2. Fermented DRB (FDRB)

DRB was fermented using *Rhizopus oryzae* (AS3.866) in a semi-solid-state system to produce fermented DRB. The fungus *Rhizopus oryzae* was acquired from the Guangdong Center for Culture Collection and Selection in China and was preserved at 4 °C on potato dextrose agar (PDA) slants. The fermentation conditions were adapted from the method outlined by Schmidt et al. with slight modifications [15]. A 250 mL conical flask containing 40 g of DRB and 20 mL of distilled water was autoclaved at 121 °C for 15 min. Once cooled to room temperature, 4 mL of *Rhizopus oryzae* spore suspension (1 × 10^5^ CFU/mL) was inoculated and thoroughly mixed with a glass rod. The mixture was then wrapped with sealing film and incubated at 30 °C for 120 h. The moisture content of the sample after drying was 6.9% (*w*/*w*).

#### 2.2.3. Fermented–Extruded Defatted Rice Bran (FEDRB)

The DRB was subjected to *Rhizopus oryzae* fermentation, dried, and subsequently treated with the extruder, following the same parameters as in Section 2.2.1 and Section 2.2.2, to prepare the FEDRB samples.

The processed rice bran was dried at 50 °C for 24 h, ground into powder, and passed through an 80-mesh sieve for further analysis.

### 2.3. Preparation of Insoluble Dietary Fiber (IDF)

IDF was prepared from DRB, EDRB, FDRB, and FEDRB using the method reported [21]. In brief, 60 g of the DRB samples was diluted at volume ratio of 1:10 with deionized water and gelatinized at 95 °C for 10 min with magnetic stirring. Amounts of 1.8 mL of α-amylase (120 KNU-S/mL), 3.0 mL of alcalase protease (4.42 AU/mL), and 1.2 mL of amyloglucosidase (300 AGU/mL) were add to the obtained DRB samples to remove starch and protein, placed in a boiling water bath for 10 min after enzymolysis, and finally centrifuged at 4000× *g* for 10 min. Then, the residue was extracted twice with hot water to obtain IDFs for different processing treatments, which were recorded as UIDF, EIDF, FIDF, and FEIDF accordingly. The basic composition of IDF was measured according to the AOAC method: moisture (930.15), protein (990.03), starch (996.11), and ash (942.05) [22,23,24,25].

### 2.4. Structural Characterization of IDF

#### 2.4.1. Scanning Electron Microscopy (SEM)

The morphology of different rice bran IDF samples was examined using a Carl Zeiss AG Merlin (Oberkochen, Germany) high-resolution field emission scanning electron microscope equipped with a secondary electron detector. The IDF samples were mounted on an SEM aluminum stub and uniformly coated with a gold layer. The coated samples were imaged (1200×magnification) under high vacuum using a low acceleration voltage of 5.0 kV.

#### 2.4.2. Fourier Transform Infrared Spectrometry (FT-IR)

We weighed approximately 5 mg of IDF samples from rice bran treated with different methods at a 1:100 (*w*/*w*) ratio. We mixed these samples with dry potassium bromide, ground them uniformly, and pressed them into transparent disks for analysis using infrared spectroscopy (VERTEX 33, Bruker, Bremen, Germany). The scanning wavelength range was 4000–400 cm^−1^, with a frequency band resolution of 4 cm^−1^, utilizing 64 scans.

#### 2.4.3. X-Ray Diffraction (XRD)

XRD analysis of the IDF samples was carried out as described. XRD patterns were acquired with a Bruker X’Pert PRO diffractometer. The following measurement parameters were used: a copper target, a radiation voltage of 40 kV, a radiation current of 40 mA, a scanning range of 5–70°, and a scanning speed of 5°/min. The relative crystallinity percentage (RCP) was determined using MDI Jade 6.5.0.32 software (Materials Data, Inc., Indianapolis, IN, USA). The integrated intensity of crystalline regions is denoted as *Icrystalline*, while *Iamorphous* represents the integrated intensity of amorphous regions.(1)$$\mathrm{RCP}=\frac{I_{crystalline}}{I_{crystalline}+I_{amorphous}}\times 100\%$$

### 2.5. Physicochemical Properties of IDF

#### 2.5.1. Water-Holding Capacity (WHC)

The method of determining WHC was that of Luo et al. [26]. A precisely weighted 1.00 g dry sample (m_0_) was hydrated with 50 mL of distilled water in a centrifuge tube at room temperature (25 °C) for 18 h with magnetic stirring. After centrifugation at 4000× *g* for 15 min, the supernatant was carefully decanted, and the residue was promptly collected and weighed (m_1_). WHC was calculated using Equation (2):(2)$$\mathrm{WHC}\;(g/g)\;=\;\frac{m_{1}-m_{0}}{m_{0}}$$

#### 2.5.2. Oil-Holding Capacity (OHC)

The OHC was evaluated using the method of Luo et al. [26]. A 1.00 g IDF sample (m_0_) was combined with 20 mL of soybean oil in a centrifuge tube and stirred magnetically at 500 rpm at room temperature (25 °C) for 18 h. The mixture was subjected to centrifugation at 4000× *g* for 15 min to decant the supernatant (free oil), and the residue was collected and weighed (m_2_). OHC was calculated according to Equation (3):(3)$$\mathrm{OHC}\;(g/g)\;=\;\frac{m_{2}-m_{0}}{m_{0}}$$

### 2.6. Functional Properties of IDF

#### 2.6.1. Sodium Cholate-Binding Capacity (SCBC)

We determined the sodium cholate-binding capacity according to the method previously described by Luo et al. [27]. Briefly, 0.5 g of each sample was incubated with 20 mL of 150 mmol/L sodium cholate solution. The mixture was agitated at 120 rpm and 37 °C for 2 h, followed by centrifugation at 4000× *g* for 15 min. We measured the residual sodium cholate content in 1.0 mL of the supernatant and calculated the adsorption amount based on the concentration difference before and after the reaction.(4)$$\mathrm{adsorption}\;\mathrm{capacity}\;(\mathrm{mg}/g)=\frac{(C_{1}-C_{2})\times 20}{m}$$
where C_1_ represents the initial concentration of sodium cholate, C_2_ denotes its concentration following adsorption, and m represents the dry weight of the sample (g).

#### 2.6.2. Cholesterol-Binding Capacity (CBC)

The CBC of the IDF samples was measured using the method employed by Luo et al. [27]. Fresh egg yolk was diluted with deionized water at a 1:10 ratio and stirred to create an emulsion. The IDF samples (2 g) were combined with 50 mL of diluted yolk emulsion. The mixture’s pH was adjusted to simulate stomach and small intestine conditions (pH 2.0 and 7.0, respectively) and incubated in a water bath at 37 °C for 2 h. Subsequently, the mixture was centrifuged at 2000× *g* for 15 min to collect the supernatants. The supernatants (1 mL) were diluted in a 1:4 volume ratio with 90% acetic acid. We utilized o-phthalaldehyde as the color-developing agent, and ascertained the cholesterol content via colorimetric analysis at a wavelength of 550 nm. The formula (Equation (5)) for calculating the amount of cholesterol adsorbed by IDF is as follows:(5)$$\mathrm{CBC}\;(\mathrm{mg}/g)\;=\;\frac{(C_{1}-C_{2})\times 50}{m}$$
where C_1_ and C_2_ denote the cholesterol concentrations in the yolk emulsion and the solution after adsorption, respectively, measured in mg/mL. The value 50 denotes the adsorption volume in milliliters, while m indicates the sample’s dry weight in grams.

#### 2.6.3. α-Amylase-Activity-Inhibitory Ability

Inhibitory capability against α-amylase activity was determined based on the methodology described by Srichamroen et al. [28]. We combined 1 g of dry sample with 40 mL of a 4% (*W*/*V*) starch solution and agitated it with 4 mg of α-amylase at room temperature for 1 h. Subsequently, we measured the glucose content in the resulting solution (C_r_). A control test (C_i_) was conducted without adding fiber. Inhibitory ability against α-amylase activity was calculated according to Equation (6):(6)$$\mathrm{Inhibitory}\;\mathrm{ability}\;\mathrm{against}\;\alpha \text{-}\mathrm{amylase}\;\mathrm{activity}\;=\;\frac{C_{i}-C_{r}}{C_{r}}\times 100\%$$

#### 2.6.4. Determination of Glucose-Adsorption Capacity (GAC)

The method for determining GAC was that of Peerajit et al. [29]. A total of 0.5 g of dry sample was mixed with 50 mL of glucose solution at concentrations of 50, 100, and 200 mmol/L and incubated in a water bath at 37 °C for 6 h to achieve adsorption equilibrium. Once adsorption equilibrium was reached, the mixture underwent centrifugation at 4000× *g* for 20 min. The adsorbed glucose, expressed in mmol per gram of IDF, was determined by the quantity of glucose solution retained by the sample, as indicated in Equation (7):(7)$$\mathrm{Glucose}\;\mathrm{adsorption}\;(\mathrm{mmol}/g)\;=\;\frac{\left(C_{1}-C_{2}\right)\times V}{M}$$
where C_1_ and C_2_ denote the initial glucose concentrations (mmol/L) and those under conditions of equilibrium, respectively; M is the dry sample weight (g); and V signifies the volume of the glucose solution (mL).

#### 2.6.5. Determination of Glucose Dialysis Retardation Index (GDRI)

In this study, GDRI is employed to forecast IDF’s capacity to slow glucose absorption in the gastrointestinal tract. According to the method of Daou et al. [30], a 0.5 g dry sample is combined with 15 mL of a 100 mmol/L glucose solution and stirred thoroughly for 1 h. The mixture is then carefully transferred to a dialysis bag (Mw = 1000). The control group consists of the glucose solution without IDF. The dialysis bag containing the sample is immersed in a 500 mL beaker with 200 mL of ultrapure water. The setup is maintained in a 37 °C constant-temperature water bath with continuous stirring for 1 h. The samples are collected at 30 and 60 min to measure glucose concentration in the dialysate. The GDRI calculation formula (Equation (8)) is as follows:(8)$$\mathrm{GDRI}\;=\;\left(1-\frac{C_{1}}{C_{2}}\right)\times 100\%$$
where C_1_ and C_2_ are the total glucose-diffused sample and the control.

### 2.7. Statistical Analysis

All experiments were repeated three times, and the data are reported in the form of mean ± standard deviation. The relative crystallinity of the IDF samples was determined using MDI Jade 6 software (MDI, Livermore, CA, USA). The analysis was conducted using SPSS 19.0 software via one-way ANOVA followed by Duncan’s multiple range test (version 19.0, SPSS Inc, Chicago, IL, USA). A statistically significant difference was identified when *p* < 0.05.

## 3. Results and Discussion

### 3.1. Basic Composition of IDFs

Table 1 presents the varied components of the rice bran IDFs obtained through different processing methods. The IDF content increased significantly by 1.98%, 5.64%, and 9.29% (*p* < 0.05) following extrusion, fermentation, and their combination, respectively. The ash contents of the other three IDF samples were higher than that of FIDF, while their starch, protein, and moisture contents were lower. The purity of the obtained IDF samples ranged from 74.11% to 77.76%, indicating that a relatively high purity was achieved.

### 3.2. Structural Properties of IDFs

#### 3.2.1. Microscopic Morphology

The scanning electron micrographs of IDF prepared from rice bran using different processing methods are displayed in Figure 1. The UIDF structure of untreated rice bran presents irregular small flakes, and the surface is dense and smooth. Compared to UIDF, EIDF and FEIDF exhibits more uneven and irregular rough cracks; they are similar in structure, composed of a large number of fragments, and the surface is wrinkled, forming larger cavities and obvious cracks, thus having a larger specific surface area. In addition, FIDF exhibits a more porous and wrinkled surface structure in comparison with UIDF. This observed disruption of the flake structure is consistent with previous reports [13,31] suggesting that such processing can lead to fiber degradation, a reduced polymerization degree, and a fragmented molecular morphology. Prior studies indicate that extrusion treatment can disrupt the rigid structure of bran cell walls, facilitating the action of native cell-wall-hydrolyzing enzymes (e.g., xylanase) to easily permeate the cell wall structure [8,32]. In fermented wheat bran, it was found that some of the cell walls in the bran were partially destroyed. The higher binding and adsorption capabilities of the processed IDF samples were attributed to increased binding-site exposure as a result of surface and microstructure modifications.

#### 3.2.2. Fourier Transform Infrared (FT-IR) Spectroscopy

Figure 2A presents the FTIR spectrum of IDF from rice bran, covering the range of 400 to 4000 cm^−1^, and illustrating its functional groups. Rice bran IDF samples had similar characteristic bands, but their absorbance and/or wavenumbers changed. The broad peak around 3400 cm^−1^ corresponds to the –OH bond in hydrogen and hydroxyl groups, primarily originating from cellulose and hemicellulose. FIDF and FEIDF show a notable blue shift in this frequency range compared to UIDF, likely due to the disruption of hydrogen-bonded organic molecules [33,34]. The results indicate that extrusion and fermentation expedite the breakdown of cellulose and hemicellulose, thereby further weakening the hydrogen bonds between their molecules. This process may expose functional groups more, altering their chemical and physical properties [35]. For example, the peak at 1640 cm^−1^ may correspond to the aromatic benzene in lignin, with the peak intensities being weaker in all processed IDF samples compared to the unprocessed one, implying a potential conversion of lignin to smaller phenolic compounds [36]. Meanwhile, the peak at 905 cm^−1^ has been identified as the characteristic region for carbohydrates bearing C-O-C and C-O-H stretching vibrations, implying the alternation of *β*-glycosidic bonds [2]. The bands observed at 2925 and 2854 cm^−1^ correspond to C-H vibrations in methyl and methylene groups. The absorption peaks of FIDF and EIDF were stronger than those of the other two, indicating that the content of soluble carbohydrates was increased here, and the two pretreatment effects were better. The band at 1655 cm^−1^ indicates a hydrogen bond between cellulose and water molecules. Compared to UIDF, the peak intensity of FIDF is weaker, indicating that the hydrogen bonds between cellulose and water molecules have been broken. These results align with the adsorption peak at 3400 cm^−1^ [13]. The band at 1160 cm^−1^ represents the characteristic peak of polysaccharides, and the peak intensity was decreased after rice bran was extruded and fermented. This may be due to the destruction of the glucoside bond in cellulose during the pretreatment of rice bran, leading to the destruction of the cellulose structure [37,38]. Consequently, extrusion and fermentation pretreatment can degrade cellulose and hemicellulose, change the microstructure of IDF, and improve its water solubility.

#### 3.2.3. X-Ray Diffraction (XRD)

The influence of extrusion and fermentation treatments on the crystallinity of IDF was examined using XRD techniques, as shown in Figure 2B. Crystalline cellulose in IDF samples was indicated by a distinct XRD peak at 20–21°. There was no significant difference in peak position among the fibers, suggesting that the crystalline structure of the fiber samples remained unchanged following different processing treatments. The crystallinity indices (Ic) were 15.52, 19.34, 16.88, and 18.27% (*p* < 0.05) for UIDF, EIDF, FIDF, and FEIDF, respectively. The notable rise in crystallinity could be attributed to the disruption of the amorphous regions during extrusion or fermentation, leading to the removal of some lignin and hemicellulose [26]. The physicochemical qualities of the fibers changed as a result of these microstructure alterations, as other researchers have previously found [13].

### 3.3. Physicochemical Properties of IDFs

The physicochemical properties (WHC and OHC) of the IDF samples are shown in Figure 3A,B. High-quality dietary fiber enhances satiety and decreases food consumption due to its superior stomach expansion and water-absorption capabilities [39]. The hydration performance of IDF is crucial to its functional and nutritional effects [40]. Figure 3 illustrates and compares the physicochemical properties of IDF, including WHC and OHC. The WHCs of the four kinds of rice bran IDF are 3.95–5.30 g/g; the WHC of EIDF is equivalent to that of UIDF, and the WHCs of FIDF and FEIDF are significantly increased by 34.18 and 13.6%, respectively (*p* < 0.05), compared with that of UIDF. Variations in the values of the four types of rice bran IDF may stem from alterations in dietary fiber structure post-processing, notably the marked rise in FIDF’s WHC. The significant increase in WHC observed for FIDF (Figure 3A) aligns well with the pronounced porous and wrinkled microstructure revealed by SEM (Figure 1). This microstructural alteration likely enhances water binding by increasing the accessible surface area. Furthermore, FTIR analysis (Figure 2A) indicated disruptions in hydrogen bonding networks (evidenced by a blue shift at ~3400 cm^−1^ and reduced intensity at ~1655 cm^−1^) [33,34,35], which may reflect the exposure of additional polar hydroxyl groups in cellulose and hemicellulose. This exposure of hydrophilic groups is hypothesized to be a major contributing factor to the elevated WHC. Previous studies [41,42] suggest that extrusion processing can enhance the water-absorption properties of wheat bran and carrot residues. However, it was noted that the extrusion processing of pea and lupin seed hulls reduced water binding [43,44]. And the WHC of rice bran IDF changes little in the extrusion process. These inconsistent results could be explained by the extrusion process parameters and the origin, content, and characteristics of the extruded fiber components [45]. The WHC of FEIDF increased more than in the extrusion treatment but less than in the fermentation treatment; this outcome might be due to the properties of extrusion and fermentation processing.

Regarding the OHC range of the four kinds of IDF at 2.04–3.64 g/g, compared with UIDF, the OHCs of EIDF, FIDF, and FEIDF increased by 36.76, 78.43, and 55.39%, respectively (*p* < 0.05). Different processing treatments notably enhanced the OHC of rice bran IDF, with fermentation showing the most pronounced effect on rice bran IDF [26]. Similarly, microfluidization treatment significantly improved the OHC of IDF from bamboo shoot shell. The enhancement in OHCs of EIDF, FIDF, FEIDF could be primarily attributed to the development of a porous structure and increased surface roughness, as clearly observed in the SEM images (Figure 1). These structural modifications are expected to increase the specific surface area and the number of potential sites available for oil binding and entrapment [26,46,47]. Through physical processing, the starch and part of the hemicellulose in the IDF can be gradually destroyed, making the IDF’s affinity to the oil group, specific surface area, and oil-binding site increase, and making its pores increase and its bulk density decrease, thus showing higher oil-holding capacity [47]. Chu et al. [46] noted that the increased OHC of the fiber after fermentation could be attributed to the enhanced contact area between the fibers and oil; this is considerably increased by the specific surface area, which, in turn, is increased by fermentation. Additionally, because more functional groups were exposed, water and oil were able to more readily bind to IDF molecules and prevent the loss of oil. Therefore, these three processing techniques can greatly increase the OHC of rice bran IDF.

The significantly enhanced WHC and OHC of processed rice bran IDF samples, particularly the FIDF and FEIDF variants, demonstrate considerable potential for precision-engineered functional foods. Their moisture retention [48] contributes to improved softness and reduced staling in bakery products, such as artisanal breads and moist cakes. Additionally, their oil-absorption capacity [49] facilitates the development of low-fat fried foods by effectively capturing frying oil in coatings, as seen in products like crispy chicken and snacks. Furthermore, these fibers help stabilize high-fat emulsions, such as mayonnaise and sausages, by inhibiting oil-phase separation [4], and serve as efficient carriers for lipid-soluble bioactive compounds [6]. Their functional properties can be specifically enhanced through processing techniques, particularly fermentation-induced OHC augmentation, enabling tailored applications across various food sectors.

### 3.4. Functional Properties of IDFs

#### 3.4.1. Sodium Cholate-Binding Capacity

The sodium cholate-binding capacity (SCBC) of dietary fiber is crucial for determining its ability to lower cholesterol. The SCBCs of IDFs from rice bran samples under different treatments are displayed in Figure 4A. The figure indicates that FEIDF’s sodium cholate-adsorption capacity is not notably different from that of UIDF. The two treatments, extrusion and fermentation, significantly improved the effect of IDF on sodium cholate. EIDF and FIDF demonstrated increases of 43.7 and 33.8% (*p* < 0.05) over UIDF, respectively, significantly enhancing the SCBS of rice bran IDF samples subjected to extrusion and fermentation. The SCBC value of rice bran IDF surpasses that of wheat bran (3.2 mg/g) and soybean bran (3.5 mg/g) [50], particularly for EIDF and FIDF, indicating its potential as a functional component for reducing cholesterol levels in vivo.

#### 3.4.2. Cholesterol-Binding Capacity

Cholesterol-binding capacity (CBC) is a key attribute of IDF, known for its potential to lower serum cholesterol levels and decrease the risk of cardiovascular disease [51]. The CBCs of all IDF samples were determined at pH = 2.0 and 7.0, respectively, to mimic the in vitro gastric and intestinal environments [52]. The enhancement of CBC through extrusion and fermentation, particularly at pH 7.0, is generally consistent with that of OHC. Figure 4B shows the adsorbed-cholesterol content of the rice bran IDF at pH 2.0 and pH 7.0. The highest CBC value of EIDF reached 7.34 mg/g (pH 2.0), and that of EFIDF reached 10.58 mg/g (pH 2.0). At pH 2.0, the cholesterol-adsorption capacity of EIDF, FIDF, and FEIDF increased by 25.7, 21, and 20.3% (*p* < 0.05), respectively, compared with UIDF, at pH 7.0. The cholesterol-adsorption capacity of EIDF, FIDF, and FEIDF was increased by 43.9, 30.8, and 47.9% (*p* < 0.05), respectively, compared with UIDF. The ability of IDF to adsorb cholesterol is related to the acidity and alkalinity of the system. This study found that rice bran IDF had a significantly higher adsorption capacity at pH 7.0 compared to pH 2.0, indicating greater cholesterol-binding capacity in the small intestine than in the stomach, consistent with findings for foxtail millet bran IDF [53] and ginseng-IDF [54]. Fermentation improves the CBC properties of millet bran DF, while extrusion significantly affects the physicochemical characteristics of DF [46]. Additionally, extrusion conditions have a significant impact on the physiochemical characteristics of IDF [1]. Qiao et al. demonstrated that fiber from extruded rice bran had a considerably higher CBC than fiber from unextruded rice bran [49]. Furthermore, in this study, the IDF treated with fermentation–extrusion had a better effect on the improvement of CBC. According to these findings, the structures of the samples were loosened by the extrusion and fermentation treatment, which increased the cholesterol-absorption capacity of rice bran IDF. The surface of the sample pores grew larger following treatment, exposing more active groups to interact with cholesterol [55,56].

#### 3.4.3. α-Amylase-Activity-Inhibitory Ability

Glucose-absorption and starch digestion are directly linked to postprandial blood glucose levels. IDF can inhibit α-amylase activity, leading to slower glucose release from starch digestion [28,57]. The α-amylase-activity-inhibition ability of IDFs from different rice bran samples was in the range of 5.32–31.37%, as shown in Figure 4C, and the α-amylase-activity-inhibition ability of the processed rice bran IDF increased. FIDF had the highest inhibitory ability against α-amylase activity (*p* < 0.05), with an inhibition rate of 31.37%, followed by FEIDF with an inhibition rate of 27.75%, and EIDF with an inhibition rate of 15.35%. The processing methods (i.e., fermentation, extrusion, combined fermentation and extrusion) significantly (*p* < 0.05) increased the initial α-amylase-inhibitory activity of IDF ∼2–5-fold. FIDF has a greater capacity to absorb glucose than EIDF, and FEIDF has a greater capacity to absorb glucose than EIDF. The α-amylase-inhibitory activity of the IDF samples was positively influenced by the different pretreatments. Our findings revealed that IDF extracted from processed rice bran demonstrated superior ability to delay glucose release from starch compared to the control group. This further demonstrated the potential of IDF in slowing the rate of glucose absorption, which was consistent with previous studies [58]. Chi-Fai [47] found that physical treatment notably reduced the bulk density of micronized fibers, exposing more alpha-amylase inhibitors to the fiber surface on the extension, embedding more starch and enzymes into the porous fiber network, and ultimately reducing glucose production. Structural modifications, such as reduced particle size, increased specific surface area, and an enhanced porous fiber network, facilitate greater embedding of starch, oil, and enzymes while exposing more α-amylase inhibitors on the fiber surface, thus reducing α-amylase activity [59].

#### 3.4.4. Glucose-Adsorption Capacity (GAC)

GAC represents the glucose-adsorption behavior of fiber during gastrointestinal transit time. The GAC of IDF powder is presented in Figure 4D. The findings demonstrate that IDF samples immersed in various glucose concentrations ranging from 50 to 200 mmol/L successfully bound glucose in amounts between 0.70 and 4.36 mmol/g. Among them, the amount of glucose bound to fiber was found to be concentration-dependent [29]. A total of 100 mmol of glucose binds IDF (*p* < 0.05) the most, and fermented IDF binds the most glucose (4.36 mmol/g), followed by fermented–extruded IDF (3.46 mmol/g). In addition, it was observed that at a high glucose concentration (200 mmol/L), glucose reached saturation and its absorption capacity decreased. Compared with untreated rice bran IDF, the three kinds of processed rice bran IDF had stronger adsorption capacity for glucose, which improved the ability of IDF to reduce postprandial blood glucose. However, the combined processing of fermented and extruded rice did not show a significant advantage over fermentation processing alone. The results of previous studies also follow this pattern, which may suggest that the effect of rice bran IDF on glucose-adsorption capacity varies depending on the processing and testing methods used [60]. Previous research indicates that both extrusion and fermentation alter the structure of fiber, and the increase in glucose-absorption capacity may be related to the changes in fiber structure following processing [13,61]. The molecules more readily penetrate the fibril as functional groups become exposed, leading to increased viscosity, porosity, and specific surface area of the fiber. This enhances the contact area between glucose and IDF, facilitating greater interaction among glucose molecules, internal molecular forces, and hydrogen bonds. Consequently, the fiber’s capacity to adsorb glucose molecules is improved, thereby reducing the glucose diffusion rate [13,58].

#### 3.4.5. Glucose Dialysis Retardation Index (GDRI)

The retardation of glucose diffusion was measured for IDF at 30 and 60 min (Figure 4E). The GDRI serves as an important in vitro predictor for assessing a fiber’s effect on slowing glucose absorption in the gastrointestinal tract [29]. The results showed that for IDF samples, after 30 min of dialysis, the GDRI values were between 7.37 and 18.44%, and after 60 min of dialysis, the GDRI values were between 16.08 and 31.97%. For all IDF samples, the 60 min GDRI value surpassed the 30 min GDRI value, the IDF value of fermentation-processed rice bran was the highest (*p* < 0.05), and the inhibition effect of glucose diffusion was better, followed by that of extrusion. The glucose dialysis delay index of IDF after processing increased significantly, indicating that the IDF functional properties of the processed rice bran were enhanced. The delay in glucose diffusion is primarily due to the adsorption capacity of the fibers and the physical barriers created by insoluble fiber particles and glucose encapsulation [48,62]. The processed IDF, characterized by a relatively high GDRI, may be attributed to its more porous surface structure (Figure 1). The fiber network structure can trap glucose molecules, slowing their diffusion [31]. Similar results were obtained by previous studies [63,64]. These findings suggest that both rice bran IDF and treated IDF samples have the ability to efficiently adsorb glucose, postpone its diffusion, and subsequently delay its absorption in the gastrointestinal system. Furthermore, IDF from fermented rice bran exhibits higher inhibition ability against GDRI and α-amylase activity, indicating potential as a functional food component to prevent diabetes [65]. Further in vivo research is needed to explore the hypoglycemic effects of these IDF samples.

### 3.5. Industrial Processing Implications

Industrial processing pathways offer distinct advantages, with extrusion demonstrating superior performance in rapid, continuous, high-throughput production, making it particularly suitable for large-scale industrial applications. Fermentation, on the other hand, uniquely enhances functional properties such as porosity and binding capacity, which are essential for the development of health-oriented formulations. An integrated approach that combines both methods can leverage synergistic effects to further optimize the structure–function relationship, although it may introduce greater operational complexity.

## 4. Conclusions

In summary, the results of this study indicated that treatment with extrusion and fermentation individually, and their combination, significantly improved the physicochemical properties of IDF by altering its microstructure, increasing its specific surface area, and promoting the exposure of functional groups. Both extrusion and fermentation disrupted the hydrogen bonds of cellulose in rice bran IDF, accelerating the degradation of cellulose and hemicellulose. FIDF outperformed the other IDF samples in WHC, OHC, GAC, GDRI, and α-amylase-inhibitory activity, while EIDF showed better CBC and SCBC. Notably, the combined FEIDF achieved the highest CBC. These findings collectively underscore the effectiveness of rice bran IDF pretreatments, particularly the combined fermentation and extrusion treatment, in enhancing its health-promoting properties. The significant improvements in key functional indices confirm the potential of processed rice bran IDF in modulation of glucose and cholesterol absorption.

## Figures and Tables

**Figure 1 foods-14-03116-f001:**
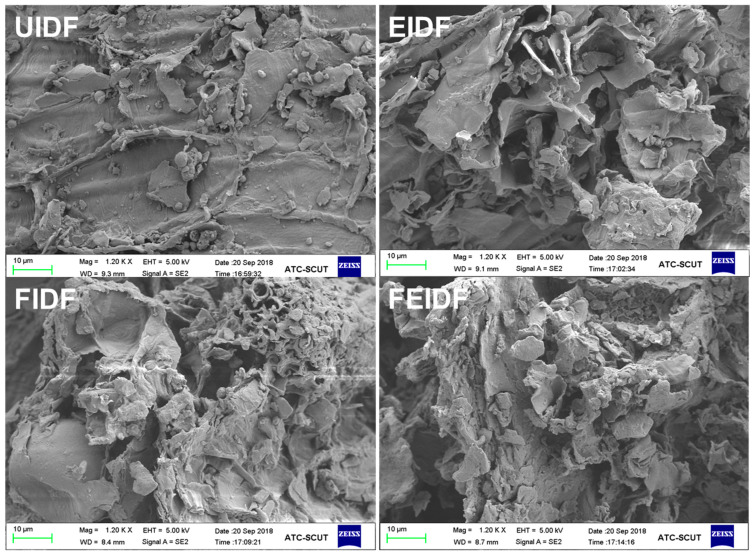
Scanning electron microscope images of IDFs from differently processed rice bran at 1200× magnification.

**Figure 2 foods-14-03116-f002:**
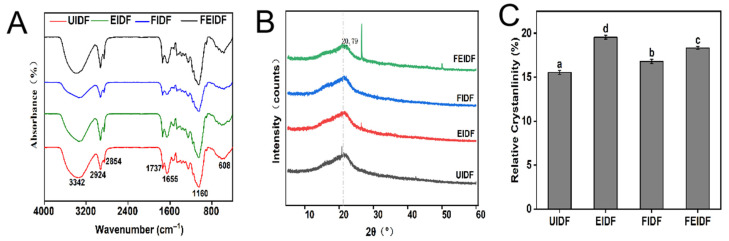
Fourier transform infrared spectroscopy (**A**), X-ray diffraction spectra (**B**), and relative crystallinity (**C**) of IDF from differently processed rice bran. Bars labeled with different letters indicate significant differences (*p* < 0.05). Values are presented as mean ± SD (*n* = 3).

**Figure 3 foods-14-03116-f003:**
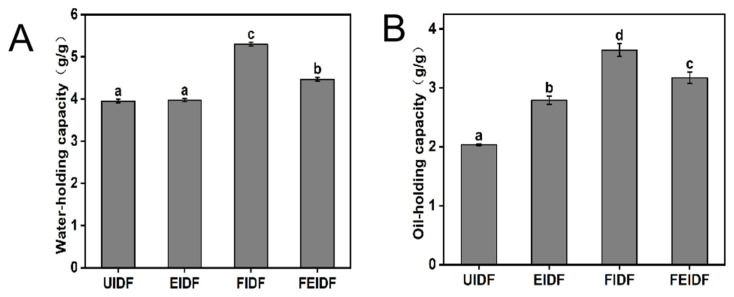
Physicochemical properties of IDF from differently processed rice bran. (**A**) Water-holding capacity and (**B**) oil-holding capacity. Bars labeled with different letters indicate significant differences (*p* < 0.05). Values are presented as mean ± SD (*n* = 3).

**Figure 4 foods-14-03116-f004:**
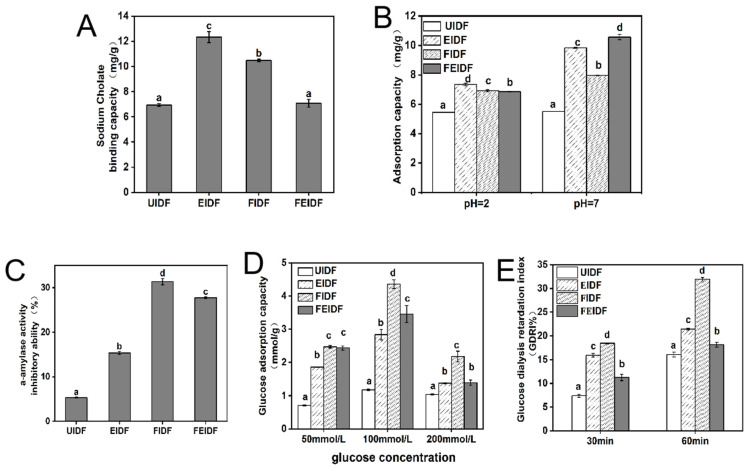
Functional properties of IDF from differently processed rice bran. (**A**) Sodium cholate-binding capacity, (**B**) cholesterol-adsorption capacity at varying pH levels, (**C**) α-amylase-activity-inhibitory ability, (**D**) glucose-adsorption capacity, (**E**) glucose dialysis retardation index. Bars labeled with distinct letters represent significant differences (*p* < 0.05).

**Table 1 foods-14-03116-t001:** The basic IDF, protein, starch, moisture, and ash contents (%) of IDFs from differently processed rice bran IDF.

	IDF Content	Protein	Starch	Moisture	Ash
UIDF	29.17 ± 0.16 a *	7.25 ± 0.06 a	4.17 ± 0.02 a	5.82 ± 0.08 c	5.13 ± 0.03 b
EIDF	29.75 ± 0.28 ab	7.69 ± 0.05 b	4.23 ± 0.05 a	4.74 ± 0.04 b	9.23 ± 0.12 d
FIDF	29.45 ± 0.28 ab	9.75 ± 0.25 d	4.67 ± 0.02 b	5.92 ± 0.08 c	3.42 ± 0.12 a
FEIDF	30.00 ± 0.26 b	7.50 ± 0.10 ab	4.24 ± 0.03 a	4.22 ± 0.04 a	6.28 ± 0.01 c

* Values in each column that do not share any letters are significantly different (*p* < 0.05). Values are presented as mean ± SD (*n* = 3).

## Data Availability

The original contributions presented in the study are included in the article, further inquiries can be directed to the corresponding author.

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
