# Peer review of "Improvement in Physicochemical and Functional Properties of Insoluble Dietary Fiber from Rice Bran Treated with Different Processing Methods"

_foods, 2025, doi:10.3390/foods14173116_

Round 1

Reviewer 1 Report

Comments and Suggestions for Authors

The article entitled” Improvement in physicochemical and functional properties of insoluble dietary fiber from rice bran treated by different processing methods” deals with the extrusion, fermentation, and combined treatment of ferment-extrusion to explore the effect of these methods on the structural, physicochemical, and functional properties of insoluble dietary fiber from rice bran. The manuscript is interesting; however, it contains several grammatical errors. Units should be consistent throughout the manuscript.

I have the following comments:

Abstract

The description of improvements (e.g., “better water-holding capacity” or “superior physicochemical and functional properties”) would benefit from including quantitative values or percentage changes to substantiate the claims.

The rice bran was degreased by petroleum ether to obtain the defatted rice bran (DRB).

Which degree of Petroleum ether was used? Which soxlet apparatus was used, a manual or an automatic one?

“According to the conditions of the laboratory's previous exploration, the extrusion parameters are set as follows, adjust the moisture content of DRB to 25%, and set the temperatures of the front, middle and rear sections of the extrusion machine to 70 °C, 98 °C and 134 °C with a screw speed of 300 r/min.” Mention the reference of the study in which these parameters are optimised for this sample.

“The processed rice bran was dried at 50°C, ground into powder, and passed through an 80-mesh sieve for further analysis.” Time duration is not mentioned in this section?

Units & Symbols – Ensure consistency in symbols and units.

“Then add heat-stable α-amylase, alcalase protease and amyloglucosidase to water bath, to remove starch and protein, boil- ing water bath for 10 min after enzymolysis, finally centrifugation at 4000×g for 10 minutes, and the residue was extracted twice with hot water to obtain IDF with different processing treatments, which were recorded as UIDF, EIDF, FIDF, and FEIDF accordingly.” The concentration of α-amylase, alcalase protease and amyloglucosidase is not mentioned?

“A precisely weighted 1g dry sample (m0) was hydrated with 50 mL of distilled water in a centrifuge tube at room temperature for 18 18 hours with magnetic stirring.” 18 is mentioned twice.

“After centrifugation at 4000 rpm for 15 minutes,” please use “x g” values, and keep it the same throughout the manuscript. Also, Use “min” throughout the manuscript.

“The OHC was evaluated using a method of Luo et al. [20]. A 1.00 g IDF sample (m0) was combined with 20 mL of soybean oil in a centrifuge tube and stirred magnetically at room temperature for 18 hours.” Mention the magnetic stirred rotation speed.

“2.6.3α-. Amylase activity inhibitory ability” Check the space

In functional property measurements (e.g., CBC at pH 2 and pH 7), please justify the choice of pH values and relate them to in vivo gastrointestinal conditions.

In 2.2.2, provide the moisture content of DRB before fermentation and confirm whether the inoculated samples were covered or aerated during incubation.

In 2.5.1 and 2.5.2, specify whether supernatant removal was performed by decanting or pipetting, as this can affect yield.

“Determination of glucose adsorption capacity(GAC)”Remove the extra space.

The FTIR result section as mentioned “The study indicates that extrusion and fermentation expedite the breakdown of cellulose and hemicellulose, thereby further weakening the hydrogen bonds between their molecules. This process may expose functional groups more, altering their chemical and physical properties” Is this change observed in the graph? If yes, please mention the peaks if they differ even slightly; there is no change in structure due to different treatments observed.

“Compared to FIDF, other three IDFs ash were higher while the starch, protein and moisture content was lower. The purity of the obtained IDF ranged from 74.11% to 77.76%, indicating that a relatively high purity of IDFS were obtained.” Check for grammatical errors.

“The sodium cholate binding capacity (SCBC) of dietary fiber was crucial in determining its ability to decrease cholesterol. Figure 4A illustrates the physical adsorption capacity of IDF from rice bran samples with different treatments , in relation to sodium cholate”. Rephrase this sentence, as it has grammatical errors.

“Cholesterol binding capacity (CBC) is a key attribute of dietary fibers , known for its potential to lower serum cholesterol levels and decrease the risk of cardiovascular disease [43]” Remove the extra space.

“Additionally, the extrusion circumstances have a significant impact on the physicochemical characteristics of dietary fibers[1],Qiao et al. demonstrated that fiber from extruded rice bran had a considerably higher CBC than fiber from unextruded rice bran[46].”
Check for the space.

“Figure 2. Fourier transform infrared spectroscopy (A ), X-ray diffraction spectra (B) and relative crystallinity (C) of IDF from different processed rice bran.”
Remove extra space.

“In summary, the results indicated that different processing treatment significantly improved IDF by altering its microstructure, increasing its specific surface area, and promoting the exposure of functional groups.”Different processing treatments were used, but in the sentence, it's mentioned a particular treatment only.

“ferment-extrusion achieved the highest CBC at pH 7” — was this pH physiologically relevant or chosen for experimental convenience?

The claim that processed IDF has potential in mitigating hyperlipidemia and hyperglycemia is promising; however, please state whether this is based solely on in-vitro results or if in-vivo validation exists, and discuss any limitations.

Author Response

Abstract

The description of improvements (e.g., “better water-holding capacity” or “superior physicochemical and functional properties”) would benefit from including quantitative values or percentage changes to substantiate the claims.

Response: Thank you for your professional review. We have carefully revised the abstract according to your recommendations (Line 14-28). Specifically, pertaining to the enhancement of the functional properties of dietary fiber in rice bran via processing treatments, we have provided comprehensive data for substantiation. The detailed description as follows: The employed processing treatments significantly enhanced the functional properties of rice bran IDF over the unprocessed sample, with increases of 1.37- to 1.78- fold in oil-holding capacity, 1.31- to 1.48- fold in cholesterol adsorption capacity, 2.89- to 5.90- fold in α - amylase inhibitory activity, and 2.41- to 3.70- fold in glucose adsorption capacity. 

The rice bran was degreased by petroleum ether to obtain the defatted rice bran (DRB). Which degree of Petroleum ether was used? Which soxlet apparatus was used, a manual or an automatic one?

Response: Thank you for your guidance. In the present work, the defatting treatment of rice bran was performed using petroleum ether with a boiling point range of 30–60 °C, and a manual extraction. The rice bran was sieved through an 80-mesh sieve and then subjected to two 24h cycles of shaking with ten times its volume of petroleum ether (30–60 °C) to remove lipids. This information has been supplemented in the resubmitted manuscript and highlighted in red text (Line 89-91).

“According to the conditions of the laboratory's previous exploration, the extrusion parameters are set as follows, adjust the moisture content of DRB to 25%, and set the temperatures of the front, middle and rear sections of the extrusion machine to 70 °C, 98 °C and 134 °C with a screw speed of 300 r/min.” Mention the reference of the study in which these parameters are optimised for this sample.

Response: Thank you for your careful review. We have supplemented the citation to the literature that informed the parameter settings used in the extrusion treatment, in the resubmitted manuscript (Line 99).

“The processed rice bran was dried at 50°C, ground into powder, and passed through an 80-mesh sieve for further analysis.” Time duration is not mentioned in this section?  Response: Thank you, the processed rice bran was dried at 50 ℃ for 24 hours. According to your suggestion, the time duration has been supplemented and highlighted in red text (Line 118).

Units & Symbols – Ensure consistency in symbols and units.

Response: Thank you for pointing out the need for consistency in symbols and units. We have carefully checked the entire manuscript and corrected all inconsistencies we found (Line 109, 112, 118, 122, 123, 127, 158, 164, 171, 172, 182, 183, 194, 201, 202, 203, 212, 216, and 217).

“Then add heat-stable α-amylase, alcalase protease and amyloglucosidase to water bath, to remove starch and protein, boil- ing water bath for 10 min after enzymolysis, finally centrifugation at 4000×g for 10 minutes, and the residue was extracted twice with hot water to obtain IDF with different processing treatments, which were recorded as UIDF, EIDF, FIDF, and FEIDF accordingly.” The concentration of α-amylase, alcalase protease and amyloglucosidase is not mentioned?

Response: Thank you. The 1.8 mL of α-amylase (120 KNU-S/mL), 3.0 mL of alcalase protease (4.42 AU/mL) and 1.2 mL of amyloglucosidase (300 AGU/mL) were add to the obtained DRB samples to remove starch and protein. In the resubmitted manuscript, these parameters have been supplemented and highlighted in red text (Line 123-126).

“A precisely weighted 1g dry sample (m0) was hydrated with 50 mL of distilled water in a centrifuge tube at room temperature for 18 18 hours with magnetic stirring.” 18 is mentioned twice.

Response: Thank you for your careful review. This sentence has been revised into "A precisely weighted 1.00 g dry sample (m0) was hydrated with 50 mL of distilled water in a centrifuge tube at room temperature for 18 h with magnetic stirring." (Line 156-158).

“After centrifugation at 4000 rpm for 15 minutes,” please use “x g” values, and keep it the same throughout the manuscript. Also, Use “min” throughout the manuscript.

Response: Thank you for your review. We agree that reporting relative centrifugal force (×g) rather than rpm is a more accurate expression, as it eliminates variability due to rotor dimensions. We also apologize for the inconsistency in reporting time units. In the revised manuscript, this issue has been addressed (Line 109, 123, 127, 165, 171, 172, 183, 202, 204 and 217).

“The OHC was evaluated using a method of Luo et al. [20]. A 1.00 g IDF sample (m0) was combined with 20 mL of soybean oil in a centrifuge tube and stirred magnetically at room temperature for 18 hours.” Mention the magnetic stirred rotation speed.

Response: Thank you, The magnetic stirred rotation speed is 500 rpm. According to your suggestion, the magnetic stirred rotation speed has been supplemented and highlighted in red text (Line 164).The OHC was evaluated using a method of Luo et al. [20]. A 1.00 g IDF sample (m0) was combined with 20 mL of soybean oil in a centrifuge tube and stirred magnetically at 500 rpm at room temperature for 18 hours.

“2.6.3 α- Amylase activity inhibitory ability” Check the space

 Response:‌We sincerely appreciate your careful review of the manuscript and your attention to the formatting details. The phrase "α-Amylase activity inhibitory ability" has been revised in text (Line 191).

In functional property measurements (e.g., CBC at pH 2 and pH 7), please justify the choice of pH values and relate them to in vivo gastrointestinal conditions.

Response: Thank you for your comment. Herein, the cholesterol binding capacity (CBC) of all IDF samples were determined at pH=2.0 and 7.0, respectively, to mimic the in vitro gastric and intestinal environments. Therefore, comparing the cholesterol absorption capacity of IDF under these two pH conditions, to some extent, reflects its binding differences in the gastrointestinal tract. In the resubmitted manuscript, this reason has been supplemented and highlighted in red text (Line 368-370).

In 2.2.2, provide the moisture content of DRB before fermentation and confirm whether the inoculated samples were covered or aerated during incubation.

Response: Thank you for your careful review. The moisture content of the sample after drying was 6.9% (w/w) and the inoculated samples were covered during incubation. According to your suggestion, we has been supplemented and highlighted in red text (Line 110-113). Once cooled to room temperature, 4 mL of Rhizopus oryzae spore suspension (1×10^5 CFU/mL) was inoculated and thoroughly mixed with a glass rod. The mixture was then wrapped with sealing film and incubated at 30°C for 120 h. The moisture content of the sample after drying was 6.9% (w/w).

In 2.5.1 and 2.5.2, specify whether supernatant removal was performed by decanting or pipetting, as this can affect yield.

Response: Thank you for your valuable comment. In Sections 2.5.1 and 2.5.2, supernatant removal was performed by decanting to ensure efficient separation while maintaining sample integrity.These revisions have been highlighted in red in the manuscript for clarity (Line 159,164-165).The supernatant was carefully decanted.

The mixture was subjected to centrifugation at 4000 rpm for 15 min to decant the supernatant (free oil).

“Determination of glucose adsorption capacity(GAC)”Remove the extra space.

 Response:‌We sincerely appreciate your careful review of the manuscript and your attention to the formatting details. The phrase "Determination of glucose adsorption capacity (GAC)" has been revised and highlighted in red text (Line 198).

The FTIR result section as mentioned “The study indicates that extrusion and fermentation expedite the breakdown of cellulose and hemicellulose, thereby further weakening the hydrogen bonds between their molecules. This process may expose functional groups more, altering their chemical and physical properties” Is this change observed in the graph? If yes, please mention the peaks if they differ even slightly; there is no change in structure due to different treatments observed.

Response: Thank you for your guidance. The fractures in the IDF structure can be verified through the results of scanning electron microscopy (SEM). In the infrared spectroscopies, slight changes in the peaks representing functional groups can be observed. The peak at 1640 cm-1 may correspond to the aromatic benzene in lignin, with peak in-tensities being weaker in all processed IDF samples compared to unprocessed one, im-plying a potential conversion of lignin into smaller phenolic compounds[33]. Meanwhile, the peak at 905 cm-1 has been identified as the characteristic region for carbohydrates bearing C-O-C and C-O-H stretching vibrations, implying that the alternation of β-glycosidic bond[34]. These results and discussion have been supplemented in the revised manuscript (Line 264-269).

“Compared to FIDF, other three IDFs ash were higher while the starch, protein and moisture content was lower. The purity of the obtained IDF ranged from 74.11% to 77.76%, indicating that a relatively high purity of IDFS were obtained.” Check for grammatical errors.

Response: Thank you for your correction. The sentence has been revised to the following, with changes highlighted in red (Line 231-234): "The ash content of the other three IDF samples were higher than that of FIDF, while their starch, protein, and moisture contents were lower. The purity of the obtained IDF samples ranged from 74.11% to 77.76%, indicating that a relatively high purity was achieved".

“The sodium cholate binding capacity (SCBC) of dietary fiber was crucial in determining its ability to decrease cholesterol. Figure 4A illustrates the physical adsorption capacity of IDF from rice bran samples with different treatments , in relation to sodium cholate”. Rephrase this sentence, as it has grammatical errors.

Response: Thank you for your correction. The sentence has been revised to the following, with changes highlighted in red (Line 355-357): "The sodium cholate binding capacity (SCBC) of dietary fiber is crucial for determining its ability to lower cholesterol. The SCBC of IDF from rice bran samples under different treatments was displayed in Figure 4A".

“Cholesterol binding capacity (CBC) is a key attribute of dietary fibers , known for its potential to lower serum cholesterol levels and decrease the risk of cardiovascular disease [43]” Remove the extra space.

Response:‌We sincerely appreciate your careful review of the manuscript and your attention to the formatting details. This sentence has been revised and highlighted in red text (Line 367-368).

“Additionally, the extrusion circumstances have a significant impact on the physicochemical characteristics of dietary fibers[1],Qiao et al. demonstrated that fiber from extruded rice bran had a considerably higher CBC than fiber from unextruded rice bran[46].”
Check for the space.

Response:‌We sincerely appreciate your careful review of the manuscript and your attention to the formatting details.This passage has already been checked for spaces and has been revised and highlighted in red text(Line 384-387).

“Figure 2. Fourier transform infrared spectroscopy (A ), X-ray diffraction spectra (B) and relative crystallinity (C) of IDF from different processed rice bran.”
Remove extra space.

Response:‌We sincerely appreciate your careful review of the manuscript and your attention to the formatting details.This sentence has had the extra spaces removed(477-478).

“In summary, the results indicated that different processing treatment significantly improved IDF by altering its microstructure, increasing its specific surface area, and promoting the exposure of functional groups.” Different processing treatments were used, but in the sentence, it's mentioned a particular treatment only.

Response: Thank you for your professional suggestion. We apologize for our ideographic opacity. The sentence has been revised into "Individual extrusion and fermentation and fermentation, and their combined treatment significantly improved the physicochemical properties of IDF by altering its microstructure, increasing its specific surface area, and promoting the exposure of functional groups", and highlighted in red text (Line 490-493).

“ferment-extrusion achieved the highest CBC at pH 7” — was this pH physiologically relevant or chosen for experimental convenience?

Response:Thank you for your careful review. This pH value is physiologically significant and is used to simulate the environment of the small intestine. Herein, the cholesterol binding capacity (CBC) of all IDF samples were determined at pH=2.0 and 7.0, respectively, to mimic the in vitro gastric and intestinal environments.In the resubmitted manuscript, this reason has been supplemented and highlighted in red text (Line 368-370).

The claim that processed IDF has potential in mitigating hyperlipidemia and hyperglycemia is promising; however, please state whether this is based solely on in-vitro results or if in-vivo validation exists, and discuss any limitations.

Response: Thank you. We agree with your opinion. In the present work, the cholesterol absorption capacity and amylase inhibitory activity of rice bran-derived IDF samples were evaluated in vitro model. We realized that the previous description regarding rice bran insoluble dietary fiber's ability to alleviate hyperlipidemia and hyperglycemia was insufficiently precise. In the resubmitted manuscript, this description has been revised as follows: The significant improvements in key functional indices confirm the potential of pro-cessed rice bran IDF in modulation of glucose and cholesterol absorption (Line 499-501).

Reviewer 2 Report

Comments and Suggestions for Authors

I have reviewed the manuscript titled “Improvement in physicochemical and functional properties of insoluble dietary fiber from rice bran treated by different processing methods” by Yanxia Chen and co-authors.

The study aimed to investigate the effects of extrusion, fermentation, and combined fermentation–extrusion on the structure, physicochemical properties, and functional characteristics of rice bran IDF, compare the relative efficiency of each processing method in enhancing health-related functional properties (lipid binding, glucose adsorption, α-amylase inhibition, etc.), providing insights for improving the utilization of rice bran as a functional food ingredient.

The novelty of the study lies in comparing three processing approaches (single and combined), in vitro, on rice bran IDF.

Here are some suggestions to strengthen the manuscript.

Abstract:

The addition of some numerical findings would strengthen the abstract.

Introduction:

The authors should state the novelty of their study.

Methodology:

  • 2.2 : I suppose the strain of Rhizopus oryzae is from the China General Microbiological Culture Collection Center (CGMCC). Please mention in the manuscript the microbial culture collection.
  • 2.3: For AOAC methods mentioned in paragraph 2.3 please add a reference.
  • 2.4.1. Scanning Electron Microscope (SEM): What were the magnifications used? It is mentioned in Figure 1, but also add here. Also, add the manufacturer’s info.
  • 2.4.2. Fourier transfer-infrared spectrometry (FT-IR): Please, indicate resolution and number of scans.
  • 2.4.3. X-ray diffraction (XRD): as described by whom? Please, provide a reference and the wavelength. Crystallinity calculation formula is needed or a reference.
  • 2.5.1. Water-holding capacity (WHC): Please specify actual temperature range.
  • 2.6. Functional properties of IDF: Please mention if standard curves for sodium cholate, cholesterol, or glucose quantification were used.
  • Statistical analysis: the authors mention significance at p < 0.05 but do not state which tests were used.

Discussion

The authors could discuss potential industrial scalability, energy cost, and environmental footprint of each method.

Author Response

Abstract:

The addition of some numerical findings would strengthen the abstract.

Response: Thank you for your professional review. We have carefully revised the abstract according to your recommendations (Line 14-28). Specifically, pertaining to the enhancement of the functional properties of dietary fiber in rice bran via processing treatments, we have provided comprehensive data for substantiation. The detailed description as follows: The employed processing treatments significantly enhanced the functional properties of rice bran IDF over the unprocessed sample, with increases of 1.37- to 1.78- fold in oil-holding capacity, 1.31- to 1.48- fold in cholesterol adsorption capacity, 2.89- to 5.90-fold in α -amylase inhibitory activity, and 2.41- to 3.70- fold in glucose adsorption capacity. 

Introduction:

The authors should state the novelty of their study.

Response: Thank you for your comment. Rice bran contains a substantial amount of insoluble dietary fiber (IDF), which possesses the efficacy to improve gut health. However, its high content of cellulose and hemicellulose affects its solubility and limits its applications. Previous studies have shown that both extrusion and fermentation with Rhizopus oryzae can promote the release of phenolic compounds from DF (including IDF), suggesting their effectiveness in modulating the structure and functional properties of the fiber. Extrusion processing can modify dietary fiber by altering physical parameters such as temperature and pressure, while fermentation with Rhizopus oryzae can change the crystallinity of cellulose and glycosidic bond linkages by releasing carbohydrate-active enzymes[19,20]. This study systematically investigates the effects of individual extrusion, Rhizopus oryzae fermentation, and their combined treatment on the structural, physicochemical, and functional properties of dietary fiber from rice bran. It preliminarily reveals the underlying patterns of physical and biological processing in modifying the functional characteristics of rice bran IDF, thereby providing a practical foundation for the processing and application of cereal-based insoluble dietary fiber. According to your suggestion, the contribution and novelty of this work have been stated in the Introduction part. (Line 72-79, 83-86)

Methodology:

2.2 : I suppose the strain of Rhizopus oryzae is from the China General Microbiological Culture Collection Center (CGMCC). Please mention in the manuscript the microbial culture collection.

Response: Response:Thank you for your guidance. The detailed description as follows: DRB was fermented using Rhizopus oryzae (AS3.866) in a semi-solid state system to produce fermented DRB. The fungus Rhizopus oryzae was acquired from the Guangdong Center for Culture Collection and Selection in China and was preserved at 4 °C on potato dextrose agar (PDA) slants (Line 105-108).

2.3: For AOAC methods mentioned in paragraph 2.3 please add a reference.

Response: Thank you for your careful review. We have supplemented the citation to the literature that AOAC methods mentioned, in the resubmitted manuscript (Line 131).

AOAC International. . Official Method 930.15: Moisture in animal feed, cereals, and oilseeds. Journal of AOAC International 2020, 103(3), 587-592.
AOAC International. Official Method 990.03: Crude protein in animal feed and plant materials. Journal of AOAC International 2020, 103(3), 593-598.

AOAC International. Official Method 996.11: Total starch in cereals and derived products. Journal of AOAC International 2025,108(2), 210-215.

AOAC International. Official Method 942.05: Total ash in food and agricultural products. Journal of AOAC International 2020, 103(3), 599-604.

2.4.1. Scanning Electron Microscope (SEM): What were the magnifications used? It is mentioned in Figure 1, but also add here. Also, add the manufacturer’s info.

Response: Thank you for your valuable suggestion. We have added the magnification factor and the manufacturer of the scanning electron microscope as follow (Line 135,138): The coated samples were imaged (1200× magnification) under high vacuum using a low acceleration voltage of 5.0 kV. The morphology of different rice bran IDF samples was examined using a Zeiss Merlin (Germany) high-resolution field emission scanning electron microscope equipped with a secondary electron detector.

2.4.2. Fourier transfer-infrared spectrometry (FT-IR): Please, indicate resolution and number of scans.

Response: Thank you for pointing out the need to indicate resolution and the number of scans.The scanning wavelength range is 4000-400 cm-1, with a frequency band resolution of 4 cm-1, utilizing 64 scans (Line 145).

2.4.3. X-ray diffraction (XRD): as described by whom? Please, provide a reference and the wavelength. Crystallinity calculation formula is needed or a reference.

Response: Thank you for your valuable comments.We appreciate the opportunity to clarify the XRD methodology and crystallinity calculation details.The following is the revised result: XRD analysis of the IDF samples were carried out as described. XRD patterns were acquired with a Bruker X’Pert PRO diffractometer, The following measurement parameters were used: copper target, radiation voltage 40 kV, radiation current 40 mA, scanning range of 5–70◦ with scanning speed of 5◦/min. The relative crystallinity percentage (RCP) was determined using MDI Jade 6.5 software (Materials Data, Inc., Indianapolis, IN, USA).The integrated intensity of crystalline regions is denoted as Icrystalline, while Iamorphous represents the integrated intensity of amorphous regions(Line147-154 ).

RCP=×100%

2.5.1. Water-holding capacity (WHC): Please specify actual temperature range.

Response: We sincerely appreciate your insightful comment.The WHC and OHC measurements were conducted under controlled temperature conditions of 25°C‌. The following is the revised result:

A precisely weighted 1.00 g dry sample (m0) was hydrated with 50 mL of distilled water in a centrifuge tube at room temperature (25℃) for 18 h with magnetic stirring (Line 159).

A 1.00 g IDF sample (m0) was combined with 20 mL of soybean oil in a centrifuge tube and stirred magnetically at room temperature (25℃) for 18 h (Line 165). 

2.6. Functional properties of IDF: Please mention if standard curves for sodium cholate, cholesterol, or glucose quantification were used.

Response: Regarding your inquiry about whether standard curves were used for the quantification of sodium cholate, cholesterol, or glucose in our study, we provide the following details:Standard curves were established for the following analytes/functionality to enable accurate quantification or evaluation:

sodium cholate quantification: y = 0.2664x + 0.0791, R2 = 0.9974

Cholesterol quantification: y = 0.246x + 0.0044, R2 = 0.9995

α - Amylase activity and glucose quantification: y = 0.7226x - 0.0356, R2 = 0.9996

Statistical analysis: the authors mention significance at p < 0.05 but do not state which tests were used.

Response:Thank you for your constructive feedback on the statistical methods used in our study. Below is our detailed response: Analysis was conducted using SPSS software by one-way ANOVA followed by Duncan’s multiple range test (version 19.0, SPSS Inc, Chicago, USA). A statistically significant difference was identified When p < 0.05 (Line 223-227).

Discussion

The authors could discuss potential industrial scalability, energy cost, and environmental footprint of each method.

Response:We sincerely appreciate the reviewer's insightful suggestion regarding the industrial applicability of the processing methods. This perspective significantly strengthens the practical relevance of our study.In direct response to this comment, In response to this comment, we have added a section to the "Results and discussion" section.

 3.5 Industrial Processing Implications

Industrial processing pathways impart distinct advantages: Extrusion excels in rapid, continuous high-throughput production, making it ideal for industrial scaling. Fermentation uniquely enhances functional properties, such as porosity and binding sites, which are critical for health-focused formulations. The combined approach leverages synergistic effects to further optimize structure-function relationships, albeit with increased operational complexity (Line 462-470).

Reviewer 3 Report

Comments and Suggestions for Authors

Please find attached the reviewers comments

Author Response

  1. In the abstract, include some quantitative or percentage improvements for the mostimportant results to give readers a sense of magnitude in your abstract section and avoid vague terms like “better”, “superior” as much as possible.

Response: Thank you for your professional review. We have carefully revised the abstract according to your recommendations (Line 14-28). Specifically, pertaining to the enhancement of the functional properties of dietary fiber in rice bran via processing treatments, we have provided comprehensive data for substantiation. The detailed description as follows: The employed processing treatments significantly enhanced the functional properties of rice bran IDF over the unprocessed sample, with increases of 1.37- to 1.78- fold in oil-holding capacity, 1.31- to 1.48- fold in cholesterol adsorption capacity, 2.89- to 5.90- fold in α - amylase inhibitory activity, and 2.41- to 3.70- fold in glucose adsorption capacity. 

2.Also, replace “furnish crucial data” with “provide important data” or “offer valuable insights”.

Response: Thank you for your valuable suggestion. We have rewrited the description of furnish crucial data as follow:These findings provide valuable insights for the development of rice bran-based functional foods with enhanced health benefits (Line 28-30) .

3.The opening statement “Dietary fiber (DF), an indigestible nutrient of organisms” is grammatically incomplete and should be restructured into a complete coherent sentence.In page 2, “Researches has confirmed…” should be “Research has confirmed…”; “lipids metabolism” should be “lipid metabolism”. Kindly go through the manuscript to correct such grammatical errors.

Response: Response: thank you for your correction. This issue has been addressed and highlighted in red . The detail description as follows: Dietary fiber (DF), an indigestible carbohydrate from plant cell walls, is called the seventh important nutrient of human (Line 35-36). Research (Line 45) has confirmed the significant health benefits associated with cereal-derived DF, including promoting intestinal health, regulation of blood glucose and lipid (Line 47) metabolism, and enhanced satiety.

  1. The manuscript switches between “dietary fibers” (plural), “dietary fiber” (singular), “DF”,and “IDF”. The authors should choose one consistent form and introduce abbreviations only once.

Response: Thank you for your valuable suggestion regarding the terminology consistency in our manuscript. We have uniformly changed "DF, IDF" to the singular form. In the following text, dietary fiber is expressed using abbreviations, and we have double-checked all of them. (Line 36, 38, 39, 42, 61, 66, 83, 179, 188, 204, 211,  339, 383, 384, 406, 408, 422, 425, 426, 427, 430, 436,456 and 461) .

  1. Key literature findings in the introduction section could be supported with data or examplesto strengthen credibility. For example, in Page 2, the statement “Recent research highlights the effectiveness…” could be supported with specific grain examples and key quantitative  Similarly, clarify the magnitude or percentage improvement, or at least the processing conditions in the sentence “enhanced release of phenolic content”.

Response: Thank you for your insightful suggestions regarding the enhancement of the Introduction section with specific data and examples. We have revised the manuscript to incorporate the following improvements:

Ren et al. (13) found that through fermentation and enzymatic treatment, the glucose adsorption capacity of dietary fiber was significantly enhanced (by 57.2% - 66.1%) (Line 67-69)

The identical extrusion and fermentation treatments ‌significantly increased‌ the phenolic content in rice bran. The total phenolic content increased by 36.3% after extrusion, and by 71.6% after fermentation, respectively ( Line 72-73)

  1. The study gap statement is repeated in parts and should be consolidated into one precisesentence directly linked to the novelty of the co mbined processing approach used in this 

Response: Thank you for your guidance. according to your suggestion, we have revised the description about the preparation of fermented-extruded defatted rice bran (FEDRB). The detail description as follows: The DRB was subjected to Rhizopus oryzae fermentation, dried, and subsequently treated by the extruder following the same parameters set in section 2.21 and 2.22, to prepare FEDRB samples (Line 116-118).

  1. In section 2.7 on statistical analysis, the statement “The relatively crystallinity of IDFsamples…..” should be rephrased for improved clarity.

Response: Thank you for your suggestion regarding the clarity of the statement in Section 2.7. We have revised the sentence to improve precision and readability: The relative crystallinity of the IDF samples were determined using MDI Jade 6 software (MDI, Livermore, CA, USA) (Line 223-224).

  1. The results and discussion section are well presented. However, many mechanisticexplanations in this section such as cell wall disruption (section 3.2.1) and exposure of polar groups (section 3.3 on physicochemical properties) are stated as fact without direct evidence from this study. It would help to clearly separate observed results from hypothesized mechanisms supported by literature. In addition, cross-reference findings from this work directly with SEM, FTIR, or XRD when attributing changes to structural modifications to strengthen internal consistency.

Response: We sincerely appreciate this constructive feedback. The following revisions address these concerns.

Both extruded treatment and fermented rice bran resulted in cracks on the surface of IDF. The flake structure was compromised, leading to fiber degradation, reduced polymerization degree, and fragmented molecular morphology[13,25].

revised: This observed disruption of the flake structure is consistent with previous reports [13,25] suggesting that such processing can lead to fiber degradation, reduced polymerization degree, and fragmented molecular morphology.

​​ "WHC of IDF by the fermentation is well in agreement with the microstructure. Thus, it reflects that the exposure of additional polar groups and other water-binding sites, particularly those associated with dietary fibers, would largely contribute to the increase in its WHC[29]. FT-IR analysis revealed that fermentation disrupted cellulose-hemicellulose connections, exposing more hydrogen bonds and dipole moments, thereby increasing WHC[35]."

revised: The significant increase in WHC observed for FIDF (Figure 3A) aligns well with the pronounced porous and wrinkled microstructure revealed by SEM (Figure 1). This microstructural alteration likely enhances water binding by increasing the accessible surface area. Furthermore, FTIR analysis (Figure 2A) indicated disruptions in hydrogen bonding networks (evidenced by blue shift at ~3400 cm⁻¹ and reduced intensity at ~1655 cm⁻¹)[30-32], may reflect the exposure of additional polar hydroxyl groups in cellulose and hemicellulose. This exposure of hydrophilic groups is hypothesized to be a major contributing factor to the elevated WHC.

 The enhancement in OHC was due to reduced particle sizes and the development of a porous structure in the modified fibers.The research indicates that the physical and chemical properties of IDF are enhanced following extruded, fermented, and fermented-extruded treatments.

revised: The enhancement in OHC of the EIDF, FIDF, FEIDF could be primarily attributed to the development of a porous structure and increased surface roughness, as clearly observed in SEM images (Figure 1). These structural modifications are expected to increase the specific surface area and the number of potential sites available for oil binding and entrapment [23,45-46].

  1. For WHC and OHC, apart from the discussion on relative changes, also discuss theirfunctional implications for food applications. Consider how these improvements can be of benefit to specific product formulations.

Response:We sincerely thank the reviewer for this valuable suggestion to deepen the discussion regarding the functional implications of the improved WHC and OHC properties. We agree that explicitly linking these physicochemical enhancements to potential food applications strengthens the practical relevance of our findings.The detailed description as follows: The significantly enhanced WHC and OHC of processed rice bran IDF samples, particularly the FIDF and FEIDF variants, demonstrate considerable potential for precision-engineered functional foods. Their  moisture retention[47] contributes to improved softness and reduced staling in bakery products, such as artisanal breads and moist cakes. Additionally, their oil absorption capacity[48] facilitates the development of low-fat fried foods by effectively capturing frying oil in coatings, as seen in products like crispy chicken and snacks. Furthermore, these fibers help stabilize high-fat emulsions, such as mayonnaise and sausages, by inhibiting oil-phase separation [4], and serve as efficient carriers for lipid-soluble bioactive compounds [6]. The functional properties can be specifically enhanced through processing techniques, particularly fermentation-induced OHC augmentation, enabling tailored applications across various food sectors (Line 342-353).

  1. When interpreting results, always ensure to report statistical significance (p-values orlettering on graphs) when stating differences as “significant” or “notably different”.

Response:Thank you for your valuable feedback on the clarity and rigor of our result interpretation.We have added significance descriptions in the manuscript results section(Line 231, 290, 304, 324,361,376,378,399,422 and 447).

  1. Kindly increase the scope of the conclusion to provide a better summary of the study.

Response: Thank you for your guidance. We agree that a more comprehensive conclusion will better highlight the significance of our study. In response to your suggestion, we have substantially revised the Conclusion section to provide a broader and more insightful summary of our work. The description as follows: In summary, the results indicated that individual extrusion and fermentation and fermentation, and their combined treatment significantly improved the physicochemical properties of IDF by altering its microstructure, increasing its specific surface area, and promoting the exposure of functional groups. Both extrusion and fermentation dis-rupted the hydrogen bonds of cellulose in rice bran IDF, accelerating the degradation of cellulose and hemicellulose. FIDF outperformed in WHC, OHC, GAC, GDRI, and α-amylase inhibitory activity, while EIDF showed better CBC and SCBC. Notably, the combined FEIDF achieved the highest CBC. These findings collectively underscore the effectiveness of rice bran IDF pretreatments, particularly combined fermentation and extrusion treatment, in enhancing its health-promoting properties. The significant improvements in key functional indices confirm the potential of processed rice bran IDF in modulation of glucose and cholesterol absorption (Line 491-502).

Round 2

Reviewer 1 Report

Comments and Suggestions for Authors

NA

Reviewer 2 Report

Comments and Suggestions for Authors

The authors have satisfactorily addressed my previous comments, and the manuscript has been improved. I consider it suitable for publication.

Reviewer 3 Report

Comments and Suggestions for Authors

The authors have utilised the reviewers comments to improve the quality of the manuscript.

It can now be accepted for publication following improvements in the quality of Figures 3 and 4.